# Towards Interpreting Visual Information Processing in Vision-Language Models

**Clement Neo**[1*]**, Luke Ong**[1]**, Philip Torr**[2]**, Mor Geva**[3]**, David Krueger**[4]**, Fazl Barez**[2,5]

[1]Nanyang Technological University  [2]University of Oxford  [3]Tel Aviv University
[4]MILA  [5]Tangentic

## Abstract

Vision-Language Models (VLMs) are powerful tools for processing and understanding text and images. We study the processing of visual tokens in the language model component of LLaVA, a prominent VLM. Our approach focuses on analyzing the localization of object information, the evolution of visual token representations across layers, and the mechanism of integrating visual information for predictions. Through ablation studies, we demonstrated that object identification accuracy drops by over 70% when object-specific tokens are removed. We observed that visual token representations become increasingly interpretable in the vocabulary space across layers, suggesting an alignment with textual tokens corresponding to image content. Finally, we found that the model extracts object information from these refined representations at the last token position for prediction, mirroring the process in text-only language models for factual association tasks. These findings provide crucial insights into how VLMs process and integrate visual information, bridging the gap between our understanding of language and vision models, and paving the way for more interpretable and controllable multimodal systems.

## 1 Introduction

Vision-Language Models (VLMs) take an image and text as input, and generate a text output. They have become powerful tools in processing and understanding text and images, enabling a wide range of applications from question answering to image captioning (Liu et al., 2023b; Li et al., 2023; Alayrac et al., 2022). Among VLM architectures, the "adapter" style has demonstrated impressive state-of-the-art performance, notable for its simplicity. These models combine a pre-trained image encoder, a pre-trained language model (LM), and a learned adapter network that maps image encoder outputs to soft prompts for LM inputs (Merullo et al., 2022; Liu et al., 2023b;a). Despite their importance and potential, the inner workings of VLMs remain poorly understood as compared to their text-only counterparts. While significant progress has been made in understanding language models (Elhage et al., 2021; Wang et al., 2023; Bricken et al., 2023) and, to a lesser extent, vision transformers (Palit et al., 2023; Vilas et al., 2023; Pan et al., 2024), there is a notable gap in our understanding of VLMs, where both modalities operate in the same space. A deeper comprehension of VLMs could lead to practical advances in safety, robustness, and functionality, similar to the progress made in language models, such as model editing (Meng et al., 2022) or identifying specific components responsible for certain behaviors (Arditi et al., 2024).

In this paper, we investigate the LM component of VLMs as they serve as the "brain" of the system, comprising more than 95% of the total parameters for models like LLaVA 1.5 7B. LLMs pre-trained on large text datasets are likely to remain the foundation of VLMs because of their powerful reasoning capabilities. Therefore, understanding how these LMs process and integrate visual information is crucial for advancing VLM interpretability.

Many open questions remain regarding the inner workings of VLMs. The nature and structure of visual representations fed into the LM remain unclear, as these visual inputs are soft prompts that

---

*Work done during ERA-Krueger AI Safety Lab internship.
Author contributions detailed in §6. Correspondence to Clement Neo <clement@clementneo.com>.

do not correspond to language tokens and cannot be interpreted as such (Lester et al., 2021; Merullo et al., 2022). This raises questions about how visual information is encoded in these representations. Furthermore, the spatial distribution of information within the image encoder's feature maps remains unclear, leaving open the question of whether and how much object-specific details are localized or dispersed across the entire representation.

Furthermore, the mechanisms by which the language model processes these visual inputs are not well understood. A significant modality gap exists between visual and textual input (Jiang et al., 2024), and it is unclear whether our mechanistic understanding of text processing in language models generalizes to the processing of visual input in VLMs.

To understand the representations in the visual inputs for VLM and how the VLM processes them, we study LLaVA 1.5 7B (Liu et al., 2023a), a popular open-source VLM with competitive performance that uses CLIP (Radford et al., 2021) as an image encoder and Vicuna 13B (Chiang et al., 2023) as its language model. We also test on LLaVA-Phi[1] and Qwen2-VL (Wang et al., 2024). We focus on a set of object identification tasks and **our main contributions** are as follows:

1. Using ablation techniques, we demonstrate that the information for an object is highly localized to the token positions corresponding to their original location in the image.

2. By extending the *logit lens* technique primarily used for language models (Nostalgebraist, 2020), we find that the representations of the visual input in the LM are refined towards the embedding of interpretable tokens in the vocabulary, despite the LM not being explicitly trained to do so.

3. By blocking the attention flow between tokens, we show that the model extracts object information from the object tokens in the middle to late layers.

Our findings are first steps in understanding the internal mechanisms of VLMs, paving the way for more interpretable and controllable multimodal systems. The code for our experiments is available at https://github.com/clemneo/llava-interp.

## 2 BACKGROUND

**Transformer Architecture.** Transformer-based autoregressive Large Language Models (LLMs) process sequences of input tokens to predict the next token in the sequence (Vaswani et al., 2017). These models achieved state-of-the-art performance across various natural language processing tasks (Brown et al., 2020), even though they are merely trained on next-token prediction (Radford et al., 2019). A LLM takes an input sequence $X = (x_1, ..., x_n)$ and outputs a probability distribution over the vocabulary $V$ to predict the next token $x_{n+1}$.

To do so, the model refines token representations layer by layer through a series of computations. Each token $x_i$ is initially represented by an embedding vector $h_i^0$ of dimension $d_m$, obtained from a lookup operation in an embedding matrix $W_E \in \mathbb{R}^{|V| \times d_m}$. This representation is then updated through successive layers using multi-head self-attention (MHSA) and feed-forward (FF) sublayers:

$$h_i^l = h_i^{l-1} + a_i^l + f_i^l \tag{1}$$

where $h_i^l$ is the representation of token $x_i$ at layer $l$, $a_i^l$ is the output from the MHSA sublayer, and $f_i^l$ is the output from the FF sublayer. All vectors $h_i^l, a_i^l, f_i^l \in \mathbb{R}^{d_m}$.

The MHSA sublayer consists of $H$ attention heads working in parallel, each implemented as follows:

$$\text{Attention}(Q, K, V) = \text{softmax}\left(\frac{QK^T}{\sqrt{d_k}} + M\right)V \tag{2}$$

where $Q, K, V \in \mathbb{R}^{n \times d_k}$ are query, key, and value matrices derived from learned linear projections of the input, and $M \in \mathbb{R}^{n \times n}$ is a causal mask. The causal mask $M$ is set to $-\infty$ in the upper right triangle, meaning $M_{ij} = 0$ if $i \geq j$, else $-\infty$. This ensures that each position can only attend to previous positions and itself, thus preserving the autoregressive property in language models. These

---

[1]Sourced from the xtuner HuggingFace model.

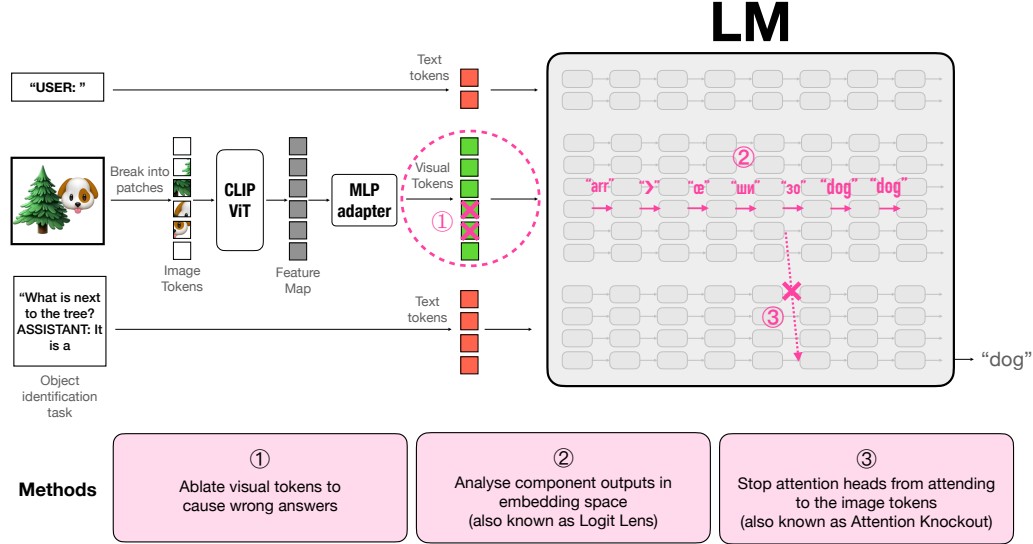

Figure 1: In adapter-style Vision-Language Models (VLMs), the visual tokens (in green) are soft prompts for the language model (LM), and are not interpretable through the vocabulary embedding. Through a set of object identification tasks, we find that (1) object information can be localized to a subset of visual tokens, (2) the representations of the visual tokens evolve towards interpretable text embeddings, and (3) the model extracts some information from the visual tokens to the last token position in the middle to late layers, to identify the object.

outputs are then concatenated and linearly projected to produce the final output of the multi-head attention sublayer.

The FF sublayer applies two linear transformations with an element-wise non-linear function $\sigma$ between them:

$$f_i^l = W_v^l \sigma(W_k^l a_i^l + b_k^l) + b_v^l, \tag{3}$$

where $W_v^l$, $W_k^l$, $b_k^l$, and $b_v^l$ are learned parameters. Finally, the representation of the last token $x_n$ at the final layer, $h_n^L$, is projected into a probability distribution over $V$ using an unembedding matrix $W_U \in \mathbb{R}^{d \times |V|}$ and a softmax operation.

**Image Encoders**. The transformer architecture has been successfully adapted for state-of-the-art image encoders, as demonstrated by Dosovitskiy et al. (2021). In this approach, input images are divided into fixed-size patches, flattened, and linearly embedded before being processed as a sequence of tokens, analogous to word embeddings in NLP tasks. Unlike text-based models, the attention mechanism in these image encoders is bidirectional (i.e., no masking is applied). The input sequence includes a special `class` token, whose final embedding is typically used for image classification tasks. The representations at the other token positions form what is called the *feature map* of the image.

Building upon this foundation, Radford et al. (2021) introduced Contrastive Language-Image Pre-training (CLIP), which jointly trains a vision transformer and a text trasnformer to produce similar embeddings for matching image-text pairs. CLIP's image encoder is currently being used for state-of-the-art performance on many downstream tasks like zero-shot image classification and vision-language models like LLaVA.

**Connecting Text and Image Models.** Tsimpoukelli et al. (2021) pioneered the concept of training an image encoder such that its output can be used by a frozen language model for multimodal few-shot reasoning. Merullo et al. (2022) advanced this approach by freezing both the pre-trained image encoder and pre-trained language model, training only a linear map that converts image features into inputs for the text model. This introduced the "adapter style" approach, utilizing pre-trained components for both modalities.

Building on this foundation, Liu et al. (2023b) introduced LLaVA, which combined a CLIP ViT-L/14 image encoder with a Vicuna-13B language model, connected by a trained linear projection, or an MLP for improved performance (Liu et al., 2023a). Crucially, LLaVA's training process focuses solely on fine-tuning for multimodal conversation, without any next-token pretraining on image-text pairs. This approach allows LLaVA to leverage the strong pre-trained capabilities of both the vision and language models while efficiently adapting to multimodal tasks.

## 2.1 NOTATION

In this section, we describe how an image and query is processed through LLaVA.

**Image Processing.** The input *image $I$* is processed by first being cropped into a square, then suitably resized before being divided into 576 image patches. The CLIP ViT-L/14 image encoder $f_{\text{CLIP}}$ processes this image to produce a *feature map $E_I = f_{\text{CLIP}}(I) \in \mathbb{R}^{N \times d_{\text{CLIP}}}$*, where $N = 576$ is the number of image patches and $d_{\text{CLIP}}$ is the dimensionality of CLIP's embeddings. An *adapter network $A$* then maps these CLIP embeddings to the language model's input space, yielding a set of *visual tokens $E_A = A(E_I) \in \mathbb{R}^{N \times d}$*, where $d_m$ is the dimensionality of the language model's input embeddings. We call $E_A$ *visual tokens* to distinguish them from the image tokens.

**Text Processing and Combined Input**. For the text input, given a tokenized prompt sequence $T = (t_1, \ldots, t_M)$, the embedding layer of the language model $E_{\text{LM}}$ maps these tokens to embeddings: $E_T = E_{\text{LM}}(T) \in \mathbb{R}^{M \times d_{\text{LM}}}$. The full input to the language model is the concatenation of the adapted image embeddings and the text embeddings: $X = [E_A; E_T] \in \mathbb{R}^{(N+M) \times d_{\text{LM}}}$, from which the language model can generate output autoregressively.

## 2.2 OPEN QUESTIONS AND HYPOTHESES

One might expect that the output of the adapter $E_A$ would produce embeddings corresponding to image information. For example, if the image contained a car, we might expect the adapter to produce tokens that correspond to the embedding of "car". However, the output of the adapter forms soft prompts that have no semantic meaning when decoded through the vocabulary (Merullo et al., 2022; Bailey et al., 2023).

### 2.2.1 LOCALIZATION OF OBJECT-SPECIFIC INFORMATION

A key question of our investigation is: **Where is object-specific information located in visual tokens?** Two conflicting hypotheses emerge from recent research:

1. **Object-Centric Localization:** Joseph & Nanda (2024) found that in classical vision transformers, token representations corresponding to specific object locations tended to align with those objects' class embeddings in the late layers, even without explicitly training for this behavior. This suggest that object information might be localized to the tokens corresponding to the spatial location of the object in the original image. However, their findings were on a vision transformer trained on ImageNet classes, and not the vision transformer in CLIP.

2. **Global Information in Register Tokens:** In contrast, Darcet et al. (2024) identified *register tokens* in vision transformers, including a CLIP variant. These are background patch tokens with unusually high norms that appear to encode global image features. It follows that the LM in an adapter-style VLM could primarily rely on these information-rich register tokens for its output.

We investigate these hypotheses in Section 3 using ablation techniques in a set of object identification tasks.

### 2.2.2 PROCESSING OF OBJECT-SPECIFIC REPRESENTATION

The processing of visual tokens may differ from text tokens. Soft prompts are very different from normal prompts (Bailey et al., 2023), and there is a *modality gap* between textual and visual representations in VLMs (Jiang et al., 2024). Furthermore, LLaVA is fine-tuned exclusively on visual question-answering (VQA) prompt-response pairs, without pre-training on next-token prediction.

This may affect how the model processes visual information as compared to text. In Section 4.1, we attempt the *logit lens* method to see how the representations of visual tokens evolve through the layers.

Recent studies have shed light on how LMs process information. Geva et al. (2023) identified a refine-then-extract process in factual association prompts, where the model first refines information at subject tokens through MLP sublayers before transferring it to the last token position via attention sublayers. Similar mechanisms have been observed in arithmetic tasks (Stolfo et al., 2023), suggesting that this (sequencing of functions) might be a general mechanism in LMs.

However, Basu et al. (2024) adapted the factual association setup to VLMs and found preliminary evidence that the LM may first summarize image information in text tokens, instead of extracting the image information to the last token position from the image tokens directly. While we do not study factual association – rather we choose to focus on a simpler object identification task, we study how information flows from the image tokens in Section 4.2.

## 3    INVESTIGATING THE REPRESENTATIONS IN MAPPED VISUAL TOKENS

In this section, we use ablation experiments to test if object information is concentrated in specific visual tokens. By ablating selected tokens and observing degradations in the model's ability to identify the object, we map how object information is distributed across the visual tokens.

**Dataset.** We use images from the COCO Detection Training set (Lin et al., 2014). To ensure the reliability of our results, we employ two filtering steps:

(1) *Choosing Simpler Images*. As a heuristic to focus on simpler images, we choose images where (a) there is an object whose size is between 1,000-2,000 square pixels, or about 2-4% of the image by area, (b) only one instance of the that object is present in the image, and (c) there are less than 4 annotated objects. We present a few examples of the images in Figure 4.

(2) *Controlling for Hallucination*. Model can sometimes hallucinate objects based on context, even when the object's visual information is removed. To address this, we create two versions of each image: the original and one where the target object is masked with noise. We only keep images where the model correctly identifies the object in the original image but fails to identify it in the masked version. This ensures that the model's object identification relies on the object's visual information.

After applying both filtering steps, our final dataset comprises 4,318 images.

**Method.** We present a high level overview of our methodology in Figure 2. Let $E_A = \{e_1, ..., e_N\}$ be the set of visual tokens. We define a subset $S \subset \{1, ..., N\}$ consisting of the indices of tokens hypothesized to contain information about a particular object $o$ in the image.

For each ablation experiment, we create a modified set of embeddings $E'_A$:

$$e'_i = \begin{cases} \bar{e} & \text{if } i \in S \\ e_i & \text{otherwise} \end{cases}$$

where $\bar{e} = \frac{1}{N}\sum_{i=1}^{N} e_i$ is the mean embedding across all visual tokens from 50,000 images from the ImageNet validation split (Deng et al., 2009). This replaces the hypothesized object-relevant tokens with an average token, effectively ablating their specific information. We do mean ablation to preserve the norm of the image token and keep them in-distribution, as their norms are typically much higher than the norm of text tokens (Bailey et al., 2023). We then evaluate the impact of token ablation using three methods:

1. **Generative Description**: We prompt the model with "Describe the image" using both the original ($E_A$) and ablated embeddings ($E'_A$). We then ascertain whether the target object $o$ is produced verbatim in the model's generated description before and after ablation. If $o$ is mentioned with $E_A$ and not $E'_A$, it indicates that the ablated tokens were crucial for the model to identify and include the object in its description.

2. **Binary Polling**: We ask the model "Is there a [o] in this image?" using both $E_A$ and $E'_A$. We then compare whether the model's next generated token changes from "Yes" to

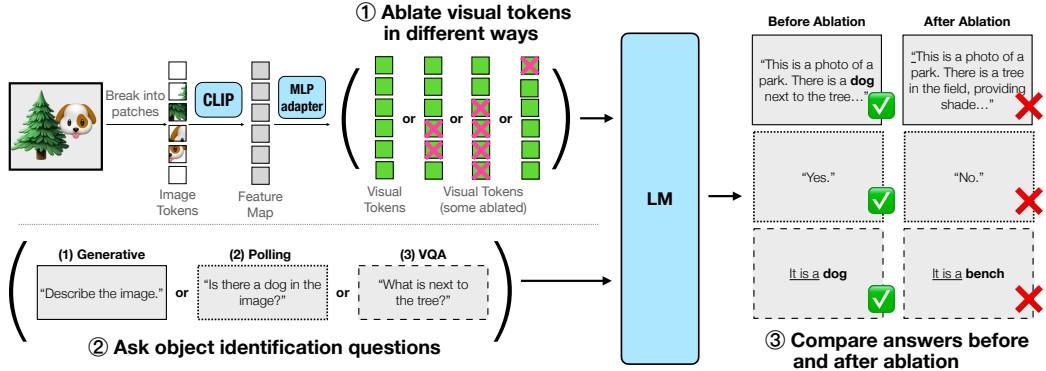

Figure 2: Overview of our ablation experiments. (1) We ablate some visual tokens that potentially contain object-specific information, (2) prompt the model to describe the image, or answer object-specific questions, then (3) measure the impact of token ablation by calculating the percentage of initially correct object identifications that become incorrect after ablation.

"No" after ablation, which indicates that the ablated tokens were crucial for identifying the object.

3. **Visual Question-Answering**: We manually curate a set of 100 images with specific questions about objects, such as "What is on the bed?". We choose questions with unambiguous answers and avoid objects directly related to the question (e.g., avoiding "pillow" for a bed-related question). We prefill the model's response with "It is a" and compare the next generated token before and after ablation. A set of examples is in Figure 5.

We choose the subset $S$ of tokens to ablate in five ways:

(1) *Object Tokens*. We choose the visual tokens whose positions correspond to the image patches that originally contained that object. (2) *Object Tokens with Buffer*. In addition to the object tokens, we include neighboring tokens. We test with two buffer sizes: 1 Buffer (including immediately adjacent tokens) and 2 Buffer (including tokens up to 2 positions away from object tokens). (3) *Register Tokens*. We select visual tokens with norms more than 2 standard deviations from the mean norm. These correspond to the register tokens identified by Darcet et al. (2024), which are thought to encode global image features. (4) *Random Tokens*. As a baseline, we ablate $n$ random tokens. (5) *High-Gradient Tokens*. As a stronger baseline, we use the Integrated Gradients attribution method (Sundararajan et al., 2017) to identify tokens most important for the model's decision. For a given object $o$, we prompt the model with "Is there a [o] in the image?" and compute integrated gradients with respect to the logit for the "Yes" token over 50 steps. We then ablate the $n$ tokens with the highest attribution scores.

**Results.** Our results are presented in Table 1. We find that ablating the object tokens significantly impairs the model's ability to identify the object. For comparable numbers of ablated tokens, object token ablation consistently results in larger performance decreases across all settings as compared to the gradient-based and random baselines. This suggests that the information about that object is localised to the region of the object token. Furthermore, these findings are consistent across all three experimental settings and applicable to both LLaVA 1.5 and LLaVA-Phi.

## 4 INVESTIGATING VISUAL INFORMATION PROCESSING

### 4.1 ANALYZING THE RESIDUAL STREAM THROUGH EMBEDDING SPACE

Can we interpret how the representations at the visual token positions evolve through the layers? To do so, we employ the *logit lens* technique (Nostalgebraist, 2020): for each layer, we decode the activation at each token position using the unembedding. Formally, for each hidden state $h_i^l$ at each token position $i$ and layer $l$, we project it into a probability distribution over $V$ using $W_U$ and take the token with the highest logit. We do this for LLaVA 1.5.

Table 1: **Performance degradation after token ablation.** A lower percentage indicates a greater impact of the ablation, meaning the model is more likely to answer incorrectly, thus suggesting that the ablated tokens contain more localized object information. Average token counts apply to both the generative and polling settings for LLaVA 1.5. We find that ablating the object tokens with adjacent tokens (33.4 tokens on average, in **bold**) significantly causes object identification performance to drop, more so than the integrated gradients and random baselines for a comparable number of tokens (40 tokens, in **bold**).

| Ablation Type | LLaVA 1.5 | | | LLaVA-Phi | | |
|---|---|---|---|---|---|---|
| *(Avg Token Count)* | **Generative** Decrease (%) | **Polling** Decrease (%) | **VQA** Decrease (%) | **Generative** Decrease (%) | **Polling** Decrease (%) | **VQA** Decrease (%) |
| Object *(12.6)* | 33.33 | 15.38 | 33.33 | 47.46 | 28.48 | 41.51 |
| + 1 Buffer (33.4) | **71.79** | **51.28** | **86.67** | **82.89** | **86.48** | **96.23** |
| + 2 Buffer (60) | 76.92 | 64.10 | 92.22 | 89.56 | 95.49 | 100.00 |
| Register tokens (3.1) | 5.13 | 0.00 | 3.33 | 7.63 | 0.00 | 1.89 |
| Int. Gradients (5) | 2.56 | 0.00 | 6.67 | 17.11 | 1.23 | 28.30 |
| Int. Gradients (10) | 20.51 | 2.56 | 6.67 | 25.26 | 4.30 | 24.53 |
| Int. Gradients (20) | 33.33 | 20.51 | 17.78 | 41.14 | 9.63 | 37.74 |
| Int. Gradients (40) | **48.72** | **35.90** | **50.00** | **61.49** | **26.64** | **56.60** |
| Int. Gradients (60) | 58.97 | 48.72 | 71.11 | 70.61 | 40.98 | 69.81 |
| Int. Gradients (100) | 76.92 | 51.28 | 84.44 | 81.23 | 61.68 | 83.02 |
| Int. Gradients (250) | 76.92 | 69.23 | 97.78 | 91.93 | 90.16 | 98.11 |
| Random (5) | 0.00 | 0.00 | 1.11 | 3.68 | 0.00 | 0.00 |
| Random (10) | 4.88 | 0.00 | 0.00 | 3.95 | 0.00 | 1.89 |
| Random (20) | 2.44 | 0.00 | 2.22 | 5.88 | 0.00 | 0.00 |
| Random (40) | **2.44** | **0.00** | **0.00** | **8.51** | **0.00** | **0.00** |
| Random (60) | 7.32 | 0.00 | 1.11 | 9.39 | 0.41 | 3.77 |
| Random (100) | 7.32 | 0.00 | 4.44 | 11.23 | 0.20 | 7.55 |
| Random (250) | 24.39 | 0.00 | 16.67 | 20.61 | 1.84 | 15.09 |

**Results.** We find that the activations in the late layers at each visual token position correspond to token embeddings that describe its original patch object in its corresponding image token. We provide some case studies in Figure 3, and an interactive demo on more examples.

To quantify this phenomenon, we analyzed 170 COCO validation images with objects of sizes between 20,000 and 30,000 square pixels (approximately 1/2 width and 1/3 height of the image). We found that in the best-performing layer for each image, an average of 23.7% of the object patch token positions correspond to the correct object class token. The best-performing layer occurs on average at layer 25.7 (out of 33 layers), confirming that mid-to-late layers tend to develop the strongest object-token correspondences (Figure 6).

We found that the tokens decoded often correspond to concepts more specific than the object-level. For example, in Figure 3a, the tokens go beyond identifying the sweater worn ("swe") and describe the specific patterns on the sweater with tokens like "diam"(ond) and "cross".

In some instances, tokens correspond to their corresponding concepts in other languages, like in Figure 3c where the tokens correspond to "year" and "month" but in Chinese and Korean instead. This is somewhat surprising given that previous work has shown that Llama used for non-English languages tend to have English tokens in their intermediate activations when interpreted through the logit lens (Wendler et al., 2024).

Overall, it is surprising that the logit lens work on VLMs! This technique works on auto-regressive LLMs because they are *pretrained on next-token prediction*, meaning their activations will be iteratively refined towards the unembedding of the predicted next token. However, VLMs are merely fine-tuned on multimodal tasks. The effectiveness of the logit lens here suggests that the hypothesis that transformers build predictions by promoting concepts in vocabulary space (Geva et al., 2022) may generalise to VLMs, even when only fine-tuned on multimodal tasks.

**Testing Other Models.** We extended our logit lens analysis to Qwen2VL-2B to verify if these findings generalize beyond the LLaVA family. Following the same methodology, we analyzed 253 COCO validation images and found that in the best-performing layer, an average of 6.5% of the visual tokens map to the correct object class token. The best-performing layer occurs at layer 25.1 (out of 29 layers), confirming the middle-to-late layer pattern observed in LLaVA (Figure 7). Inter-

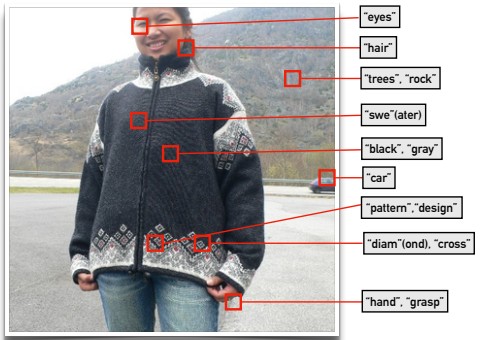
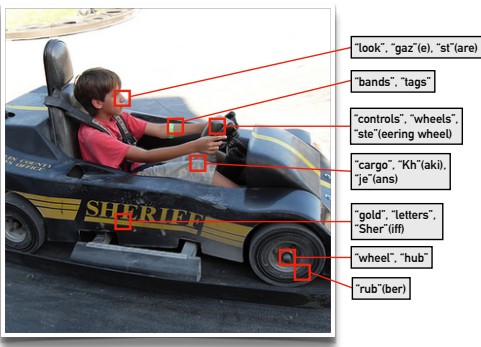

(a) An image of a lady in the sweater. The logit lens identifies tokens that correspond to specific detail of the sweater, such as "pattern" and "diam"(ond).

(b) An image of a child in a go-kart. The representations sometimes encode specific details, such as "look" and "gaz"(e) instead of just "face".

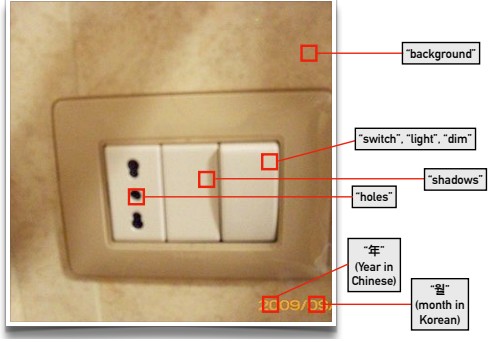
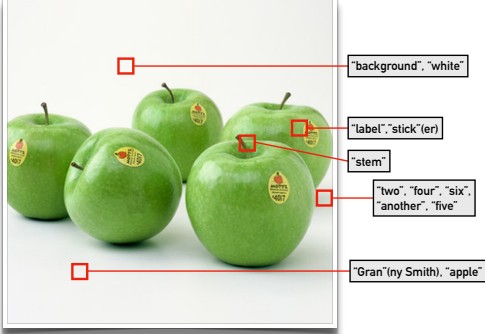

(c) An image of a switch. In the intermediate layers, the year and month tokens are encoded in non-English characters.

(d) An image of a bunch of apples. Global features, like count, show up in background tokens, though this may be an artifact of the LM processing.

Figure 3: Examples of tokens and their positions that the logit lens yields in the late layers. The labelled tokens come from a range of layers around the middle-to-late layers. In our labelling for an image, tokens that are part of a word are completed with an educated guess e.g. "diam"(ond)

estingly, Qwen2VL shows non-zero correspondence even in early layers, possibly due to its cross-attention adapter providing better initial mapping compared to LLaVA's simpler MLP adapter. Both models demonstrate increasingly aligned token representations with object tokens in later layers, suggesting this refinement pattern is a general property of adapter-style VLMs.

**Case Study: Global Features.** We sometimes find global-level features, such as numbers possibly referring to object counts, appearing in unexpected background tokens (Figure 3d). Ablating these specific tokens doesn't change the model's description, and the numbers persist in the logit lens even after ablation. This suggests these global features may be artifacts of the language model's processing rather than direct information from visual tokens—potentially explaining why CLIP-based VLMs often underperform CLIP itself on classification tasks (Zhang et al., 2024).

**Potential Applications**. This may be useful for getting coarse segmentation maps for little to no extra computation, which could lead to downstream applications. For example, Liu et al. (2024) suggests to adjust the attention weights towards the visual tokens in the LM to reduce hallucinations. We speculate that directing attention towards specific objects may improve this method, though we leave further exploration to future work.

## 4.2 TRACING ATTENTION FLOW

To understand how visual information is processed and integrated within the language model, we investigate the flow of critical information through the network. Specifically, we examine whether the

model directly extracts information from relevant visual tokens, or whether it aggregates information from visual tokens before transferring it to the final output token.

**Method.** We employ the *attention knockout* technique introduced by Geva et al. (2023). This method involves selectively blocking attention between specific tokens at different layers of the model, allowing us to assess the importance of various connections for the model to identify the object. Our process is as follows:

(1) For a given input and target layer $\ell$, we artificially block specific attention connections in the multi-head self-attention (MHSA) sublayer by setting the corresponding attention mask values to negative infinity: $M_{rc}^{\ell+1,j} = -\infty \quad \forall j \in [1, H]$ where $M^{\ell+1,j}$ is the attention mask for head $j$ at layer $\ell + 1$, and $r$ and $c$ are the indices of the tokens between which we want to block attention.

(2) We apply this blocking over a window of consecutive layers, including early layers (L1-10), early to middle layers (L5-14), middle layers (L11-20), middle to late layers (L15-24), and late layers (L21-31).

(3) For each window, we measure the decrease in accuracy and prediction probability for the correct token when blocking attention under three scenarios: (i) From object tokens and their surrounding buffer to the final token. (ii) From all non-object tokens to the final token. (iii) From all visual tokens except the last row to the last row of visual tokens[2].

We conduct this experiment using LLaVA 1.5 on our curated Visual Question Answering (VQA) dataset of 100 images.

**Findings.** Our results are presented in Table 2.

We find that blocking attention from the object tokens (and their buffers) to the final token in mid-late layers leads to noticeable performance degradation. This suggests that the model directly extracts object-specific information in these later stages. Blocking attention from non-object tokens to the final token in early layers also causes some performance drop. This indicates that contextual information from the broader image is processed and integrated in the earlier stages of the model.

Interestingly, blocking attention from visual tokens to the last row of visual tokens has minimal effect on performance. This contrasts with findings by Basu et al. (2024), who suggested that the model might summarize image information in the last row of visual tokens before using it. Our results indicate that, at least for our object identification tasks, the model doesn't rely on such a summarization step.

Table 2: Results of blocking attention between different token groups across various layers of the LLaVA 1.5 model. The values represent the relative performance on the correct token being predicted when blocking attention, with 1.00 being no impact and 0.00 being all questions wrong post-ablation.

| Attention Blocking | | Layer | | | | | |
|---|---|---|---|---|---|---|---|
| From[*] | To[†] | Early | Early-Mid | Mid | Mid-Late | Late | All |
| O | LTP | 1.00 | 1.00 | 0.96 | 0.88 | 0.93 | 0.82 |
| O+1 | LTP | 1.00 | 0.99 | 0.90 | 0.82 | 0.89 | 0.67 |
| O+2 | LTP | 1.00 | 1.00 | 0.91 | 0.80 | 0.91 | 0.68 |
| I-(O+1) | LTP | 0.88 | 1.00 | 0.97 | 0.96 | 0.98 | 0.82 |
| O+1 | LVR | 1.00 | 1.00 | 1.00 | 1.00 | 1.00 | 1.00 |
| I-LVR | LVR | 1.00 | 1.00 | 1.00 | 1.00 | 1.00 | 1.00 |

[*]**From:** O = Object Tokens, O+$n$ = O + $n$ Buffer,
I-(O+1) = All visual tokens except O+1, I-LVR = All visual tokens except last row
[†]**To:** LTP = Last Token Position, LVR = Last Visual Token Row

---

[2]We do this as recent work by Basu et al. (2024) suggests that the model may summarise visual information in the last row of visual tokens, though they study this in a different VQA task. If the model does use the last row of visual tokens as a 'working summary', then blocking the attention from all the visual tokens to the last row should prevent it from answering the question.

**Testing Other Models.** To verify that our attention blocking results are not specific to LLaVA-1.5, we conducted additional experiments with LLaVA-Phi-3. Rather than using fixed layer groupings, we employed a sliding window of 10 layers to obtain finer-grained results across the network. We find that the results (Figure 8) closely mirror our original results: blocking attention from object tokens to the last token position shows the strongest effect in middle-to-late layers, while blocking attention from visual tokens to the last row continues to show minimal impact.

## 5  RELATED WORK

**Explainability.** Classical techniques like GRAD-Cam (Selvaraju et al., 2017) and Integrated Gradients (Sundararajan et al., 2017), focus on input attribution, while recent tools like LVLM-Interpret (Stan et al., 2024) extend these methods to VLMs. Our work complements these by investigating the model's internal processing mechanisms.

**Mechanistic Interpretability (MI) for LLMs.** Techniques such as causal tracing (Meng et al., 2022) and sparse coding (Bricken et al., 2023; Huben et al., 2023; Marks et al., 2024) have advanced our understanding of how LLMs perform tasks despite being primarily trained for next-token prediction, but VLMs are not trained on next-token prediction for visual inputs (Liu et al., 2023b). Our work extends some of these techniques to understand visual information processing in multimodal contexts.

**MI for Multimodal Models**. In CLIP, researchers have discovered multimodal neurons (Goh et al., 2021), decomposed image representations (Gandelsman et al., 2023; 2024), identified interpretable network subgraphs (Rajaram et al., 2024), and extracted interpretable features using sparse coding (Rao et al., 2024; Daujotas, 2024; Fry, 2024). In contrast, our work focuses on generative VLMs.

For generative VLMs, Schwettmann et al. (2023) found neurons corresponding to visual concepts, while Palit et al. (2023) adapted causal tracing for BLIP. While these studies help us understand the role of specific components in the VLM, we investigate the representations of the visual inputs.

Basu et al. (2024) used causal tracing and attention contributions in a visual question-answering setup to find that VLMs such as LLaVA retrieve information from earlier causal layers of the LM (i.e. layers 1-4 vs layers 4-7 in an LLM) when answering a multi-modal question. Their analysis suggests that the visual information may be summarised in a consistent subset of late visual tokens. In contrast, we study VLMs in a more straightforward object-identification setting. We also examine their information transfer hypothesis through our attention blocking experiments in Section 4.2.

**Parallel Work**. Concurrent research by Jiang et al. (2025) independently arrived at similar logit lens findings and developed a hallucination reduction method based on these insights. Wu et al. (2024) observed comparable logit lens patterns across both visual and audio modalities.

## 6  CONCLUSION

Our investigation into visual information processing in LLaVA reveals that object information is highly localized within visual tokens corresponding to the object's spatial location, challenging the hypothesis that VLMs primarily rely on global features in register tokens. Visual token representations evolve to align with interpretable textual concepts across layers, suggesting VLMs refine visual information toward language-like representations even without next-token prediction pretraining for visual inputs. Our attention blocking experiments suggest that VLMs may extract object information from relevant visual tokens in mid-to-late layers. These findings bridge the gap between our understanding of language and vision models, providing a foundation for more interpretable multimodal systems.

**Limitations & Future Work**. We focused on LLaVA-type models as a starting point for interpreting more complex VLMs, but our results may not generalize to significantly different architectures. Our study concentrates on object identification tasks, providing a simpler setup compared to multi-step reasoning. While we tested object identification in three ways, our findings may not fully represent VLM behavior in complex tasks like reasoning or open-ended question answering. Future work should explore practical applications in reducing hallucinations and enhancing factual accuracy, while examining whether these findings extend to broader task types and model architectures.

## AUTHOR CONTRIBUTIONS

Clement Neo conceived the study, designed and performed the experiments, and led the writing of the manuscript. Luke Ong contributed to the writing and literature review. Philip Torr, Mor Geva and David Krueger provided advice and comments. Fazl Barez served as the primary advisor for this work and helped significantly with the design and write-up of the paper.

## ACKNOWLEDGMENTS

We thank the Torr Vision Group for providing compute and hosting Clement Neo during the internship.

We thank Sarah Schwettmann, Ashkan Khakzar, William Rudman, Joseph Miller, Alex Spies for discussing and reading drafts of the paper.

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

# A  FILTERED DATASET EXAMPLES

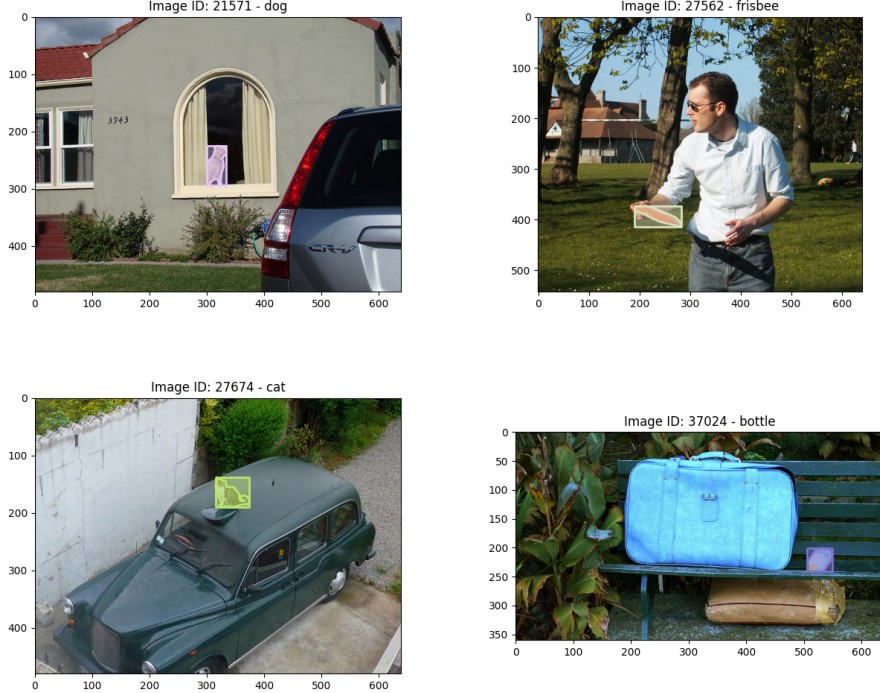

Figure 4: Example of dataset images for the object identification tasks.

## B  CURATED IMAGES AND QUESTIONS EXAMPLES

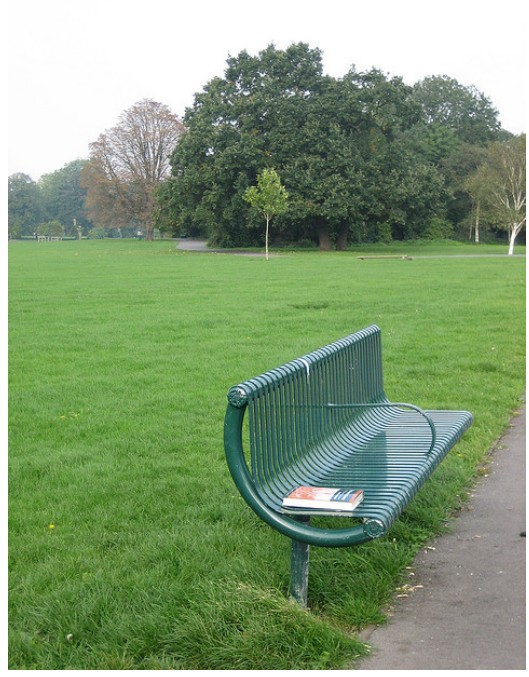

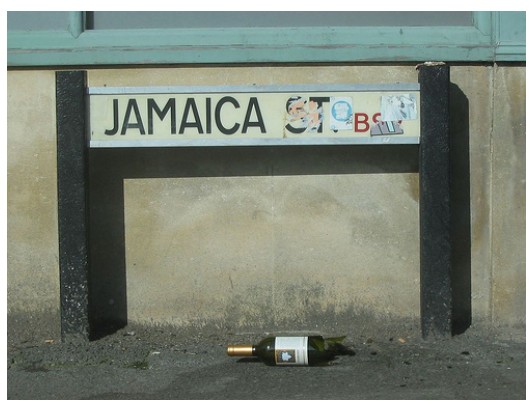

(a) "What is on the bench? It is a" → book          (b) "What is below the street sign? It is a" → bottle

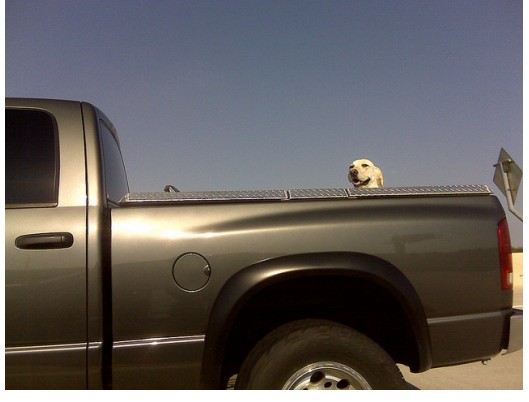

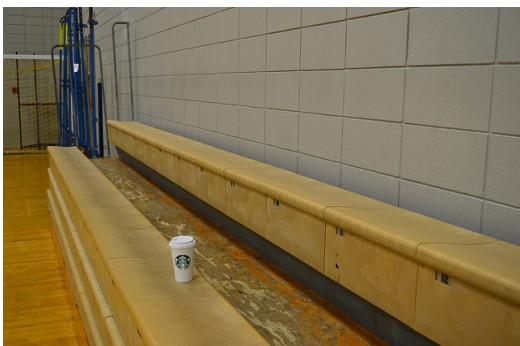

(c) "What is on the truck? It is a" → dog          (d) "What is on the seat? It is a" → cup

Figure 5: Examples of curated images and their questions. We ask the model the question and prefill its answer with "It is a". The correct answer is underlined.

## C  OBJECT CORRESPONDENCES

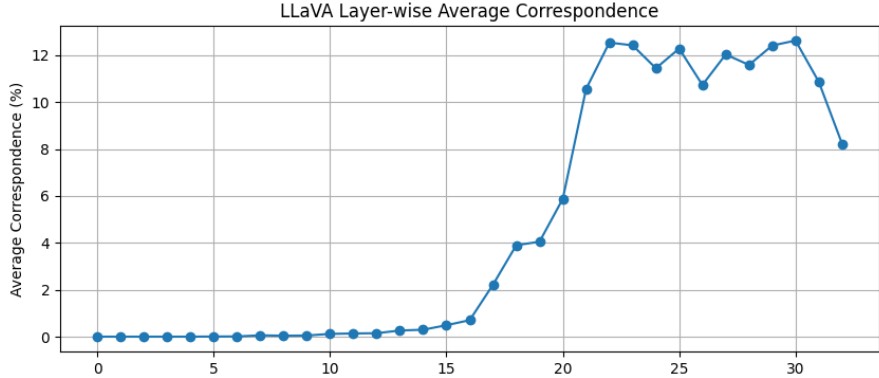

Figure 6: Token-to-object correspondence in LLaVA-1.5 across transformer layers. Analysis of 170 COCO images shows peak correspondence (23.7%) occurs in later layers (average layer 25.7/33).

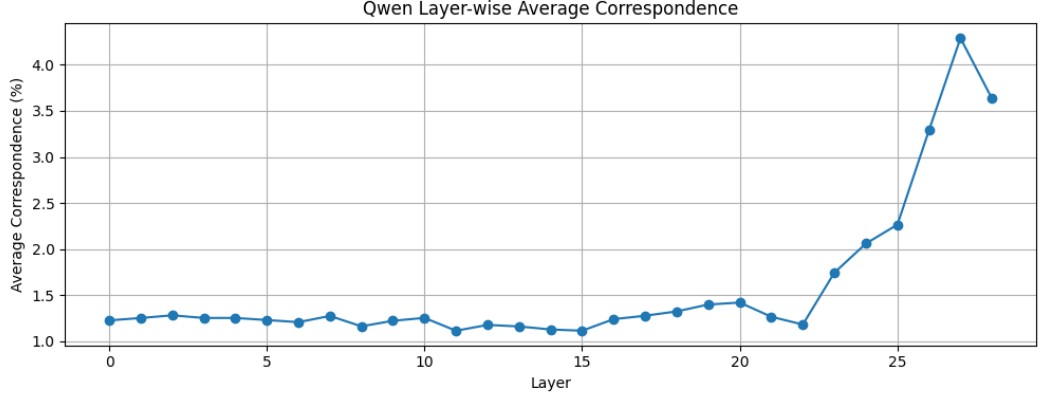

Figure 7: Token-to-object correspondence in Qwen2VL-2B showing increasing alignment in deeper layers, peaking at layer 25.1/29 with 6.5% correct mapping. Unlike LLaVA, Qwen exhibits non-zero correspondence in early layers.

# D  LLAVA-PHI-3 BLOCKING

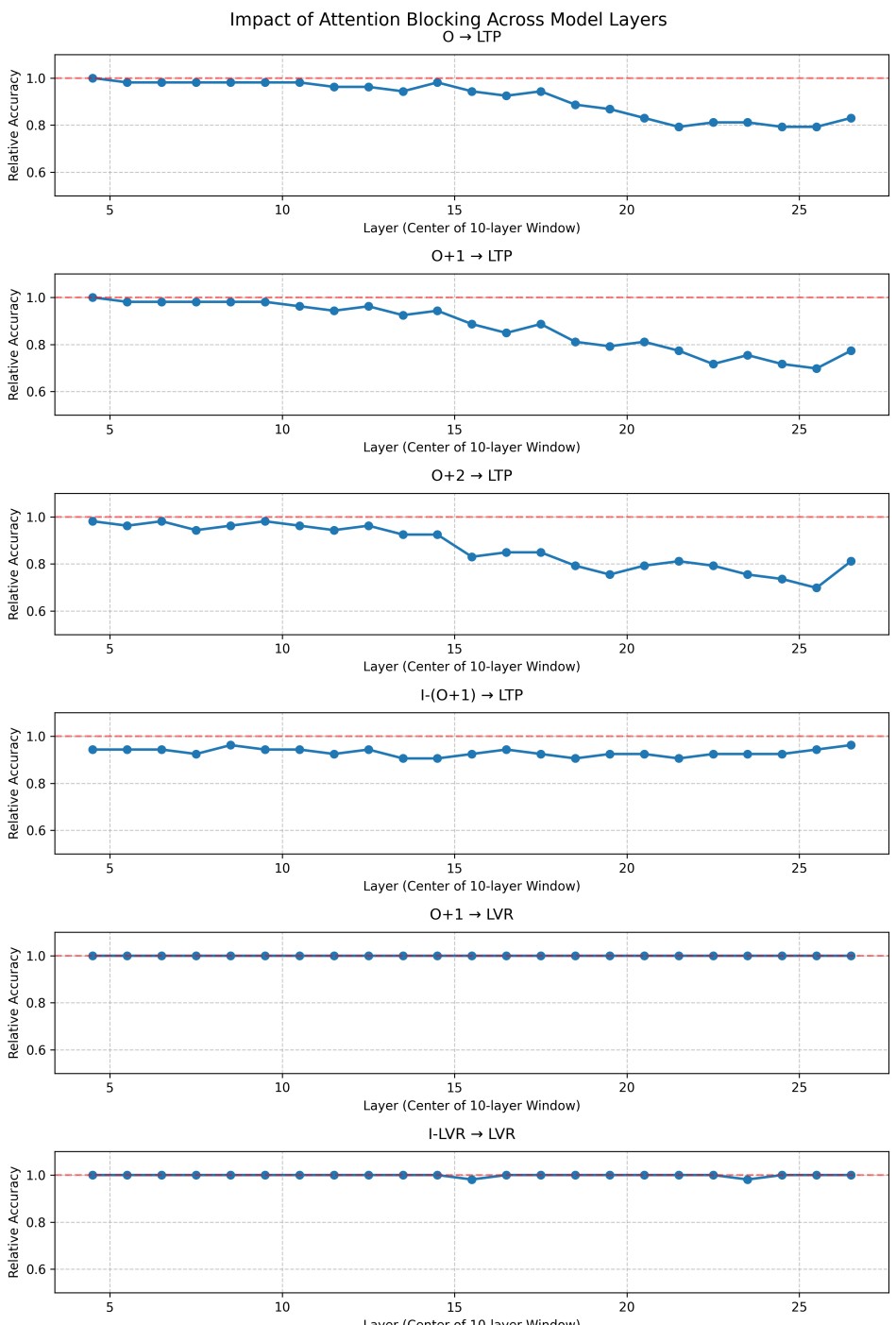

Figure 8: Attention blocking results for LLaVA-Phi-3 using a sliding window of 10 layers. We find that similar to LLaVA-1.5, blocking attention from object tokens to the last token position has the strongest effect in middle-to-late layers, while blocking attention from visual tokens to the last visual row shows minimal impact throughout the network.

