# OpenReview forum: "Towards Interpreting Visual Information Processing in Vision-Language Models"
_ICLR.cc/2025/Conference — ICLR 2025 Poster_

### Official Review · Reviewer_LYyP · 2024-11-04

**Soundness:** 3
**Presentation:** 2
**Contribution:** 3
**Rating:** 6
**Confidence:** 3

**Summary:**

This work proposes some interpretable experiments to investigate the flow of visual information processing in LLaVA-based VLMs. The authors find 3 synthetic tasks to test whether the blurred object tokens will affect the prediction accuracy of VLMs. Secondly, the attention blocking experiments investigate whether the VLMs rely on visual information summarization to predict the correct answer. The findings are interesting and vital to VLM development.

**Strengths:**

1. This work provides very important insights to the visual information processing of VLMs. Even if previous phenomena like the correlation between high-resolution SFT and good performance in LLaVA-1.6 has too some degree revealed that the object visual tokens can be very significant to the correct image understanding of VLMs, the quantitive analysis on such effects are not conducted.

**Weaknesses:**

1. The contributions on the interpretability of visual information processing in VLMs are well recognized, while the contributions on how we can adopt such information processing pattern on better prompting or training for VLMs are limited. For example, if the object visual tokens are vital for the answer prediction, is it possible that simply repeat them or augment them can lead to significant performance improvement? And maybe some text prompts or soft-prompts can well utilize such object visual tokens for better model understanding?

2. The investigations focus on object only, which are only important to the general VQA, captioning, and some specific object detection tasks. Beyond object visual tokens, the OCR tokens, text rendering tokens, and knowledge-related tokens can also be your study object. For example, what is the effect of blocking the important text tokens to the tasks of TextVQA, OCRVQA, MME, POPE?

**Questions:**

1. The definition of last row of visual tokens in line 435 is very confused to me. For example, CLIP-L has 16*16 visual tokens, does it mean the last [-1,;] 16 tokens? It is recommend to provide a lot of explanations on that and why such experiments on last row visual tokens are required?

2. The experiments in Table 2 is highly related to how you split the early/mid/late layers. Since Phi-3 has less layers than the LLM used in LLaVA, vicuna-7b, is there any difference on your findings when you adopt LLaVA-Phi-3 as investigation VLMs?

---

> ### Author Response · Authors · 2024-11-21
> **Response 1**
>
> We thank the reviewer for their comments on the paper, as well as their suggestions on how to improve the paper.
>
> ---
>
> **The contributions on the interpretability of visual information processing in VLMs are well recognized, while the contributions on how we can adopt such information processing pattern on better prompting or training for VLMs are limited.**
>
> ---
>
> We thank the reviewer for recognising the contributions we make towards the interpretability of VLMs. Regarding how these findings could be used for better prompting or training of VLMs, we did suggest that this could be used to improve the hallucination rate for VLMs by modifying the Paying More Attention to Image methodology by Liu et al. (2024) [1] to pay more attention towards _object_ tokens, now that we know that the object information is quite local to the visual tokens that originally corresponded to the object.
>
> In general, our focus was more towards rigorously showing that the object information is local and is refined towards interpretable tokens. Wendler et al (2023) [2] suggests that LMs think predominantly in their main language regardless of the input language; we think our results suggest a more general phenomenon where LMs think predominantly in their main language regardless of the input modality. We hence think that our findings have larger implications overall on understanding multimodality, and focused on understanding the model’s processing.
>
> [1] https://arxiv.org/abs/2407.21771
> [2] https://arxiv.org/abs/2402.10588
>
> ---
>
> **The investigations focus on object only, which are only important to the general VQA, captioning, and some specific object detection tasks.**
>
> ---
>
> The key investigation of our study is on the visual token set — on how they are structured, how they are processed and how their information is extracted. Hence, we focus exclusively on object identification because it gives us the clearest setting in which the required information to complete the task is almost exclusively in the visual tokens.
>
> We note that Basu et al. (2024) [3] study some setting closer to the textVQA setup, which makes it a much more complicated process because it requires extracting some intrinsic knowledge about the LM. (e.g. they show the model a picture of Christopher Nolan, and expect the model to answer “what movies has this director made?”.) This confounds the information flow because now the information comes from both the processing of the visual information (Who is this person?) and intrinsic knowledge from inside the LM (what movies has Christopher Nolan made?). Hence we think studying the object identification setting is the clearest for understanding visual token processing, and we focus on this.
>
> ---
>
> **The definition of last row of visual tokens in line 435 is very confused to me... It is recommend to provide a lot of explanations on that and why such experiments on last row visual tokens are required?**
>
> ---
>
> Yes, if CLIP ViT-L/14 has 24*24 tokens, the last row visual tokens refer to the last 24 tokens. This is because Basu et al. (2024) [3] suggest that visual information could be summarised in the last row of visual tokens, because they found that only the last row token is involved in transferring information to the last token. However, they do so merely by looking at the attention pattern in the early layers (Section 4.2 of their paper). By doing attention blocking from the entire image to the last visual token row across all layers, we demonstrate that even in the most extreme case there is no performance degradation, suggesting that looking at the attention patterns may not be enough, or that their findings may not generalise past the factual VQA task they study.
>
> We noted that this was not stated outside of the findings section, and should have been mentioned in the Method section to clearly give a motivation on why we do so. We thank the reviewer for pointing this out. We will add the following paragraph to the methods section in the Camera Ready version of the paper at line 435:
>
> “...(iii) From all visual tokens except the last row to the last row of visual tokens.
>
> We do this as recent work by Basu et al. (2024) suggests that the model may summarise visual information in the last row of visual tokens, though they study this in a different VQA task. If the model does use the last row of visual tokens as a ‘working summary’, then blocking the attention from all the visual tokens to the last row should prevent it from answering the question.”
>
> [3]  https://arxiv.org/abs/2406.04236

---

> > ### Author Response · Authors · 2024-11-21
> > **Response 2**
> >
> > ---
> >
> > **The experiments in Table 2 is highly related to how you split the early/mid/late layers... Since Phi-3 has less layers than the LLM used in LLaVA, vicuna-7b, is there any difference on your findings when you adopt LLaVA-Phi-3 as investigation VLMs?**
> >
> > ---
> > While we agree that the set of layers to do attention blocking for is a crucial experiment parameter, we point out that we do try to cover for the possible variances by also including the early-to-mid layers and the mid-to-late layers.
> >
> > Nevertheless, we try the experiments with LLaVA-Phi-3, doing the same attention blocking experiment, this time with a sliding window of size 10, as opposed to simply choosing a set of early/mid/late layers. We report the results in the graph [here](https://i.imgur.com/mNyQHuV.png). This graph is comparable to the results in Table 2 of the paper, but with much more variation in the range of layers covered.
> >
> > We find that the overall trend of the attention blocking having more of an effect in the middle-to-late layers also happen in LLaVA-Phi-3! For the instances where we block the attention from the object to the last token position (O -> LTP, O+1 -> LTP, O+2 -> LTP), we see a dip in accuracy in the middle to late layers, before it slightly recovers in the late layers. We also see that blocking the attention from the visual tokens to the last row (O+1 -> LVR, I-LVR -> LVR) also has no significant effect on the model. This mirrors our findings with LLaVA and further strengthens our findings by doing attention blocking over a more comprehensive range of layers.
> >
> > ---
> >
> > We hope that we have answered your questions satisfactorily. We note that our main focus of the paper was on the interpretability of the model, and we show rigorously that the information in the visual tokens is highly local, and that the representation of these tokens are refined towards interpretable text tokens. We think that this may be useful for applications, such as reducing hallucinations though we leave this to future work.
> >
> > As you feel that our contributions to the interpretability of the model was well recognised, we hope to have addressed your concerns on the interpretability methods, and we would kindly ask you to adjust your review score while taking the rebuttal into account. We’d be happy to answer any further questions you might have.

---

> ### Author Response · Authors · 2024-11-25
> **Follow-up**
>
> Once again, thank you for your feedback! Since the discussion period closes soon (in 1-2 days), please let us know if you have any questions about our response! We feel that we have addressed your concerns in our response, and would like to know if you have any further concerns. If you have no other questions and felt we have addressed your concerns, we would kindly ask you to consider increasing your score!

---

> > ### Comment · Reviewer_LYyP · 2024-11-28
> >
> > Thanks for the explanations on the last of visual tokens and the discussions on adopting Llava-phi-3. I raised the score to 6.

---

### Official Review · Reviewer_nkbz · 2024-11-04

**Soundness:** 3
**Presentation:** 4
**Contribution:** 3
**Rating:** 8
**Confidence:** 4

**Summary:**

This paper presents a study of the processing dynamics of visual tokens within adapter-style VLMs, particularly focused on investigating where object information may be stored and how it is used throughout the model.

The study is anchored around 3 main methods:

1. The ablation of tokens pertaining to the locations of objects within images (by replacing with a globally averaged token). This is compared against the ablation of register tokens, random tokens, and high-gradient tokens (determined using the Integrated Gradients method for selecting tokens most influential in a given model decision). Experimental results (Table 1) show that  the ablation of object level tokens results in a significantly higher degradation in performance when compared against ablating comparable amounts of random or high-gradient tokens.

2. Applying a Logit Lens style analysis to intermediate layers' representations to qualitatively study what information is encoded in intermediate representations of tokens as they flow through the model. A (I believe qualitative) study over this shows that activations of visual tokens in later layers of the model tend to be decoded by the unembedding matrix into tokens pertaining to concepts related to the image (e.g. an "eye" on a face).


3. Attention knockout to better understand the importance of certain token group dyads for model performance. Results show that "knocking-out" attention from object tokens to the last visual token (which is conjectured to be used by VLMs to reason about outputs) results in a greater performance degradation when compared against knocking out all visual tokens except for those in the last row to the visual tokens in the last row.

**Recommendation**
I think the paper is generally very well written, easy to understand, and generally supports its claims about the locality of object-level information to tokens pertaining to object regions in images. I have a few questions and concerns (see Weaknesses and Questions below) which I would value the authors addressing before I finalize my score (and which I think addressing could help strengthen the clarity of the paper). I am generally leaning towards recommending acceptance.

**Strengths:**

**Presentation**: The paper is very clearly written and the provided figures do an excellent job of communicating the experimental setup and results. I have some questions about implementation details that appear omitted, but everything that was included in the paper was generally very easy to follow due to the quality of the writing.

**Token Ablation Experiments**: The experiments presented in section 3 appear quite thorough and soundly motivated/constructed. The filtration strategy selected to simplify the experimental setup makes sense, and the selection of baselines to compare object token ablation against provides good coverage of alternative explanations to the papers' hypothesis. As a result, the provided experimental results appear to clearly show the presence of object information within the tokens of object regions in images.

**Weaknesses:**

**Attention Flow Experiments**: The attention flow experiments appear somewhat inconclusive. The provided experimental results do indeed show a distinct performance degradation when knocking out attention from object tokens in later layers to the last visual token when compared against knocking out attention from all rows to the last row. However, the authors state in L219-L228 that there are competing explanations about where information is aggregated (e.g. just within visual tokens vs. also within text tokens). I would have therefore expected an additional condition in Table 2 showing the performance of knocking out some variant of text token attention. I would appreciate it if the authors could comment on this.

**Missing Details**: This is minor, but I think there's a number of experimental details missing whose inclusion could stand to strengthen both the clarity and reproducibility of the paper (see Questions below).

**Questions:**

* L257 - 259: This is likely not necessary to clear up, but I'm curious if the authors tried ablating tokens by zeroing out embeddings rather than using the global average token? I imagine this might just generally hurt performance potentially adding a confounding factor to the experiments. Depending on the answer, perhaps mentioning the reason behind this design choice could add clarity.

* L262-L266: How is it ascertained whether the target object o is mentioned in the generated description? Is this done manually or automatically?

* L242-L246: How is correct identification determined? Is this automated or manual?

* L329-:330: I'm wondering what the precise definition of "comparable number of tokens" is here? Is this compared to the average number of object tokens per image in the dataset?

* L350-L352: How are correspondences identified, is this done by hand? If so, could the authors please comment on the statistics of how many images were evaluated, and if some measure of percentage of correspondences found was measured? In other words, I'm trying to get a sense of how reliably this behavior is observed in the model. Given the current level of detail provided, readers may assume that these results were cherry picked.

* L369: "such numbers seemingly counting applies in background tokens." -- the wording is a little awkward, consider revising.

---

> ### Author Response · Authors · 2024-11-21
>
> We thank the reviewer for their review and appreciate their thorough feedback on the paper.
>
> ---
>
> **The attention flow experiments appear somewhat inconclusive... I would have therefore expected an additional condition in Table 2 showing the performance of knocking out some variant of text token attention.**
>
> ---
> We note that the competing explanation was for another experimental setting, and we were checking if that explanation would apply to our case as well.
>
> In the factual association framework, it covers cases where the model is asked a question like:
> **“Beats Music is owned by ___”**
> Where implicit information needs to be extracted from the transformer. In this case, it is the information that Apple is the owner of Beats Music. In this case, the model refines the information at the last subject token position “sic”.
>
> In our case, the tokens that need to be refined and information extracted from are from the visual token set. This is even more so as we explicitly try to avoid objects that are directly related to the question, e.g. If we ask the model “What is on the bed?” We only ask that question if the object is not directly related to the bed (e.g. book, as opposed to pillow). Hence, our hypothesis is more of a “does the refine-then-extract mechanism in the factual association framework also apply for the object identification setting?“ for the attention blocking experiment we do not consider the text.
>
> We note that the setup more similar to Geva’s setup we were referring to was by Basu et al. (2024) [1], where they gave the model a photo of Christopher Nolan and asked the model “What movie by this director won a Golden Globe?” In this case, they were able to find that the model used the MLP at the position of “this director”’.
>
> In contrast, we maintain that our study focuses on the visual token set, and how the model processes them. This is why the attention blocking experiment is in a setup with less confounding by the text, so that it makes sense to only block attention from the object tokens.
>
> Nevertheless, since the attention blocking worked less well than expected, it could be that the model does do some sort of aggregation at the token. But since our setup has questions of different formats (e.g. what is on the bed? What is to the left of the bus? What is the child holding?), this would necessitate a different experimental setup where the text prompts are more fixed. Hence, we leave further exploration of text token blocking to future work.
>
> [1] https://arxiv.org/abs/2406.04236
>
>
> ## Questions
> ---
>
> **L257 - 259: ... I'm curious if the authors tried ablating tokens by zeroing out embeddings rather than using the global average token.**
>
> ---
>
> After doing some exploratory analysis, we found that the image tokens form soft prompts that are of high magnitude (~10-80, as compared to ~0-1 for text tokens), which is generally the case for soft prompts. For this reason we felt that to do zero-embedding would push the input further out of distribution, and it would be best to do mean-ablation. We agree that mentioning this will add clarity to the design choice, and will include this in the camera-ready version of the paper.
>
> ---
>
> **L262-L266: How is it ascertained whether the target object o is mentioned in the generated description? Is this done manually or automatically? ... L242-L246: How is correct identification determined? Is this automated or manual?**
>
> ---
>
> In both cases, `o` is detected automatically based on whether class is in description, through a simple `class_name in description` detection. We found that this worked well enough, especially because COCO class names are sufficiently general.
>
> ---
>
> **L329-:330: I'm wondering what the precise definition of "comparable number of tokens" is here?**
>
> ---
>
> Yes, the average number of object tokens is used. To be specific, we were comparing the average number of object tokens ablated for +1 buffer (which is 33.4), and comparing them to the integrated gradients and random baselines where 40 tokens were ablated.
>
> ---
>
> **L350-L352: How are correspondences identified, is this done by hand?... I'm trying to get a sense of how reliably this behavior is observed in the model.**
>
> ---
>
> This was done by hand in the main paper, but we have done additional experiments to make sure that this behavior can indeed be reliably detected. Please refer to the general response for more details.
>
> ---
>
> **L369: "such numbers seemingly counting applies in background tokens." -- the wording is a little awkward, consider revising.**
>
> ---
>
> Thank you for the suggestion, we will revise this to ‘such as number tokens that could be referring to the number of apples in the images’.

---

> ### Author Response · Authors · 2024-11-25
> **Follow-up**
>
> Once again, thank you for your feedback! Since the discussion period closes soon (in 1-2 days), please let us know if you have any questions about our response!

---

> > ### Comment · Reviewer_nkbz · 2024-11-28
> >
> > Thank you to the authors for the kind reminder and for the thorough response to my questions and concerns. My concerns have largely been addressed and so I am raising my score to 8.

---

### Official Review · Reviewer_c4uP · 2024-11-07

**Soundness:** 4
**Presentation:** 3
**Contribution:** 3
**Rating:** 6
**Confidence:** 4

**Summary:**

This paper studies how VLMs, particularly LLaVA, process and integrate visual information. Through ablation studies, the authors show that object information is localized in specific visual tokens, which evolve into interpretable text-like representations across layers. Attention flow experiments reveal that the model extracts object-specific information in mid-to-late layers while processing contextual information in earlier layers.

**Strengths:**

1. The paper is well-written and well organized. There are effective diagrams and explanations that make complex concepts accessible.
2. The paper presents an innovative approach to understanding how visual tokens are processed in the language model component of VLMs. The insights on visual token alignment could inform practical techniques to improve VLM robustness and reduce hallucinations in multimodal tasks.
3. The authors design comprehensive experiments and ablation studies to support the findings.

**Weaknesses:**

1. The experiments only focus on the LLaVA model. The authors can consider other prevailing VLMs.
2. The authors can provide more direct evidence (experiments) to support the potential applications of the proposed methods.

**Questions:**

Is it easy to apply the methods and findings in this paper to downstream tasks?

---

> ### Author Response · Authors · 2024-11-21
>
> We thank the reviewer for their feedback on the paper. We are pleased that the reviewer finds that our paper is well written with comprehensive experiments.
>
> ---
>
> **The experiments only focus on the LLaVA model. The authors can consider other prevailing VLMs.**
>
> ---
> We thank the reviewer for suggesting that we should try our analysis on other models.
>
> We have tried performing an adapted version of the logit lens on InstructBLIP and Qwen2VL, and have found that the visual tokens become refined towards interpretable tokens in the residual stream as well. Please refer to the general response for more details.
>
> ---
>
> **The authors can provide more direct evidence (experiments) to support the potential applications of the proposed methods... Is it easy to apply the methods and findings in this paper to downstream tasks?**
>
> ---
>
> The methods and findings from this paper have limited direct applicability to downstream tasks in their current form. We do suggest one potential application - using our insights about object token localization to modify Liu et al.'s (2024) 'Paying More Attention to Image' [1] methodology to potentially reduce VLM hallucinations.
>
> Our research was primarily focused on understanding the fundamental mechanisms of how VLMs process information, specifically showing that object information is localized to specific visual tokens and gets refined into interpretable representations. We think that our findings concretely contribute to the broader theoretical understanding of multimodal processing, and we leave further research into practical applications to future work.
>
> [1] https://arxiv.org/abs/2407.21771

---

> ### Author Response · Authors · 2024-11-25
> **Follow-up**
>
> Once again, thank you for your feedback! Since the discussion period closes soon (in 1-2 days), please let us know if you have any questions about our response!

---

### Official Review · Reviewer_UGCm · 2024-11-07

**Soundness:** 4
**Presentation:** 4
**Contribution:** 3
**Rating:** 8
**Confidence:** 3

**Summary:**

This work applies mechanistic interpretability techniques to understand how visual information is localized and processed in VLMs. This work:

- Tries to localize where information processing happens within VLM transformers
- Applies the “logit lens” (Nostalgebreist 2020) to understand how representations of visual tokens evolve across layers of the transformers
- Applies attention knockout to see how blocking attention between different subsets of tokens in the transformer causes the accuracy of the VLM to decrease

For localization, they find that ablating object tokens with a buffer has the most impact on model performance on the 3 ablations, with two LLaVA VLMs. For the logit lens, they find that patch embeddings at different layers can be mapped to corresponding meaninful tokens in the tokenizer vocabulary.

**EDIT 11/25/2024:** I am raising my score from 6 to 8, and soundness from 3 to 4, after the author response.

**Strengths:**

1. Important problem: The authors address an important problem of understanding how visual knowledge is localized and processed in transformer-based VLMs.

2. The formulation of each sub-question is well-motivated, the method and metrics are well-designed, the analysis of each sub-question is really thorough, and the findings are well summarized.

3. The paper is very well-written and easy to follow.

**Weaknesses:**

1. **Evaluated insufficient VLMs:** As the authors acknowledged, the analysis is performed on only two VLMs from the same (LLaVA) family. However, the methodologies should be easily extendable to other transformer-based VLMs.

    A key sticking point of mechanistic interpretability methods is that we do not know how the findings generalize to other models, so it is hard to take any findings at face value when they have only been performed on one model (especially when some of the findings contradict findings from other works, e.g. VLMs rely on features stored in register tokens).

    I suggest including findings from other VLMs such as InstructBLIP, QwenVL, InternVL, and seeing if your findings generalize to more models.

2. **Results of "logit lens" analysis are hard to interpret:** While the authors find some interesting mappings between visual patches and output tokens, it's hard to tell if these are meaningful correspondences or an artifact of cherry-picking/confirmation bias, since we see mappings for only a handful of patches, and only at one layer (although it is unclear which one).

    One way to make this analysis seem less cherry-picked is to annotate patch-token pairings manually for whether they are a "matching" pair or not, and analyze what % of patches map to a matching token. You could also look at how these mappings evolve across layers.

**Questions:**

In Figure 3, what layer do these unembeddings correspond to? Are these all from a specific layer, or did you select instances from different layers that matched human interpretations?

---

> ### Author Response · Authors · 2024-11-21
>
> We thank the reviewer for their feedback on the paper. We are pleased that the reviewer finds that the paper is well written and clearly addresses an important problem in the field of multimodal interpretability.
>
> ---
>
> **Evaluated insufficient VLMs: As the authors acknowledged, the analysis is performed on only two VLMs from the same (LLaVA) family. However, the methodologies should be easily extendable to other transformer-based VLMs.**
>
> ---
>
> We thank the reviewer for suggesting that we should try our analysis on models of other families. We have tried performing an adapted version of the logit lens on Qwen2VL, and have found that the visual tokens become refined towards interpretable tokens in the residual stream towards the late layer as well. Please refer to the overall response for more details.
>
> ---
>
> **Results of "logit lens" analysis are hard to interpret...**
>
> ---
>
> We thank the reviewer for pointing out that the logit lens findings could be an artifact of confirmation bias, and for suggesting concrete examples of further analysis.
>
> We have done additional experiments to make our logit lens analysis more concrete. Please refer to the overall response for more details.
>
> We also encourage the reviewer to look at the interactive demo provided in the paper (and [here](https://llava-logit-lens.netlify.app/) for easy access).
>
> ---
>
> **Question: In Figure 3, what layer do these unembeddings correspond to?..."**
>
> ---
>
> These unembeddings are from different layers that are in the middle-to-late layer range. We agree that this may have been ambiguous and will update the caption to be more specific, to (added sentence in bold):
> “
> Examples of tokens and their positions that the logit lens yields in the late layers. **The labelled tokens come from a range of layers around the middle-to-late layers.** In our labelling for an image, tokens that are part of a word are completed with an educated guess e.g. “diam”(ond).
> “

---

> ### Author Response · Authors · 2024-11-25
> **Follow-up**
>
> Once again, thank you for your feedback! Since the discussion period closes soon (in 1-2 days), please let us know if you have any questions about our response!

---

> > ### Comment · Reviewer_UGCm · 2024-11-25
> > **Response to Rebuttal**
> >
> > Apologies for the late response. Thank you for the additional experimental details; the quantitative results and qualitative demos are quite compelling. I will increase my rating from 6 to 8.

---

### Author Response · Authors · 2024-11-21
**General Response**

We thank all the reviewers for their thorough review and useful feedback on the paper. We are pleased that all reviewers find the paper well-written and its contribution towards the interpretability of VLMs comprehensive and well-supported. Here, we want to address a few common concerns that some reviewers brought up:

---

**Concern 1. How concrete are the logit lens findings? Are they cherry picked?**

---

Qualitatively, we find that models do quite often refine objects and background scenes towards interpretable tokens through the logit lens! We encourage reviewers to take a look at the [interactive demo](https://llava-logit-lens.netlify.app/), which includes the full logit lens for not just the figures in the paper, but also 100 randomly picked images. We think that just a few seconds of scrolling would show that this phenomenon of the tokens being refined towards their corresponding object to be quite consistent, which reasonably supports our claims.

Nevertheless, qualitatively, we set out to ascertain how consistently this happens as suggested by some reviewers. We first note a potential complication: If we look at just the top token of each patch, the top token might be too descriptive, e.g. in the sweater example in Figure 3(a) where the tokens of the diamond pattern on the sweater refine towards “diam”(ond) and “cross” rather than “swe”(ater). So in this example we won’t expect to see the “swe”(ater) token to appear across all the tokens, but we should expect to see it appear for a proportion of it. [1]

We take the COCO validation set and filter for images with objects of sizes between 20,000 and 30,000 square pixels. This roughly corresponds to an object of size about 1/2 width and 1/3 height of the image. This results in 170 images. For each image and its associated object, we analyze the token predictions at each patch position corresponding to the object's location across all layers of the model. We find that:
* In the best-performing layer for each image, an average of 23.7% of the object patch token positions corresponds to the correct object class token.
* The best-performing layer occurs on average at layer 25.7 (out of 33 layers) suggesting that mid-to-late layers in the model tend to develop the strongest object-token correspondences.

A graph of the average correspondence for each layer across all images for Llava is available [here](https://i.imgur.com/3lXuIqK.png).

---

**Concern 2. The paper should try exploring models outside of the LLaVA family.**

---

We have tried performing an adapted version of the logit lens on Qwen2VL-2B, and have found that the visual tokens become refined towards interpretable tokens in the residual stream as well.

We filter the COCO validation set in the same way as above, resulting in about 253 images. The number is higher as LLaVA crops the images to a square, while Qwen2VL doesn’t. We do the same experiment and find that:
* In the best-performing layer for each image, an average of 6.5% of the visual tokens map to the correct object class token. Note that this number is likely lower because for LLaVa we are taking the proportion of the object tokens, while for Qwen2VL we are taking the proportion of all visual tokens since the size of the visual token set for Qwen2VL varies.
* The best-performing layer occurs on average at layer 25.1 (out of 29 layers), which is also the middle-to-late layer range for Qwen2VL.

A graph of the average correspondence for each layer across all images for Qwen2VL is available [here](https://i.imgur.com/vqac2rO.png).

We see across both models that the token representations become more aligned with the object token representation as we go later in the layers. Interestingly, for Qwen2VL the correspondence is non-zero in the early layers, which could be due to its better ability to map the visual tokens better to the language embedding space using its cross-attention adapter (as compared to just 1-layer MLP for Llava.)

**Ablation Experiments.** We note that the ablation experiments may not apply for other families as they do not preserve the visual token set (e.g. InstructBLIP uses an auxiliary network to process the feature map). Nevertheless, we believe that our ablation experiments on LLaVA are enough to suggest that the upstream image encoder, CLIP, encodes the information in its feature map in a strongly local manner. Since many VLMs still use CLIP as its encoder, we believe that this finding will better enable further studies on how other VLMs use auxiliary networks to process CLIP’s feature maps.

[1] One further complication is that if the object is too general, it won’t show up in the logit lens at all. So for the LLaVA experiment here we had to manually filter out the ‘person’ class because it never showed up in the logit lens, because it would often predict more specific details like ‘face’ or ‘arm’

---

### Meta-Review · Area_Chair_Htom · 2024-12-09

**Metareview:**

This paper applies several mechanistic interpretability techniques to understand how visual information is localized and processed in VLMs such as LLaVA. After rebuttal, it received scores of 6688. All the reviewers are happy about the paper, commenting that the paper is well written, with comprehensive experiments. The well-designed ablation studies and thorough analysis are helpful to understand VLM information processing. Given all the positive scores this paper has received, the AC would like to recommend acceptance of the paper.

**Additional Comments On Reviewer Discussion:**

Three out of the 4 reviewers have increased the scores after author rebuttal. Specifically,

1. Reviewers have asked the author to extend experiments beyond LLaVA. During rebuttal, the authors have provided additional results on Qwen2-VL, with similar findings.

2. Reviewers have also shown concerns regarding the representativeness of logit-lens findings, and questions about attention flow methodology, the authors have also provided detailed response to address reviewers' concerns.

3. There are also some limitations of the work. For example, it's unclear how the method here can be applied beyond object visual tokens, such as OCR and knowledge-related tokens, as knowledge benchmarks and text-rich image understanding benchmarks are also very popular ways to evaluate multimodal LLMs.

---

### Decision · Program_Chairs · 2025-01-22

Accept (Poster)